# Integrated Analysis of Transcriptome and Metabolome Reveals Molecular Mechanisms of Rice with Different Salinity Tolerances

**DOI:** 10.3390/plants12193359

**Published:** 2023-09-22

**Authors:** Zhenling Zhou, Juan Liu, Wenna Meng, Zhiguang Sun, Yiluo Tan, Yan Liu, Mingpu Tan, Baoxiang Wang, Jianchang Yang

**Affiliations:** 1Jiangsu Key Laboratory of Crop Genetics and Physiology/Jiangsu Key Laboratory of Crop Cultivation and Physiology, Agricultural College, Yangzhou University, Yangzhou 225009, China; zhouzl13716@163.com; 2Lianyungang Academy of Agricultural Sciences, Lianyungang 222000, China; zhiguangsun@126.com (Z.S.); 15805132001@163.com (Y.T.); ly516.bester@163.com (Y.L.); 3College of Life Sciences, Nanjing Agricultural University, Nanjing 210095, China; 2021116018@stu.njau.edu.cn (J.L.); 2022116038@stu.njau.edu.cn (W.M.); tempo@njau.edu.cn (M.T.)

**Keywords:** transcriptome, metabolome, salt stress, rice

## Abstract

Rice is a crucial global food crop, but it lacks a natural tolerance to high salt levels, resulting in significant yield reductions. To gain a comprehensive understanding of the molecular mechanisms underlying rice’s salt tolerance, further research is required. In this study, the transcriptomic and metabolomic differences between the salt-tolerant rice variety Lianjian5 (TLJIAN) and the salt-sensitive rice variety Huajing5 (HJING) were examined. Transcriptome analysis revealed 1518 differentially expressed genes (DEGs), including 46 previously reported salt-tolerance-related genes. Notably, most of the differentially expressed transcription factors, such as NAC, WRKY, MYB, and EREBP, were upregulated in the salt-tolerant rice. Metabolome analysis identified 42 differentially accumulated metabolites (DAMs) that were upregulated in TLJIAN, including flavonoids, pyrocatechol, lignans, lipids, and trehalose-6-phosphate, whereas the majority of organic acids were downregulated in TLJIAN. The interaction network of 29 differentially expressed transporter genes and 19 upregulated metabolites showed a positive correlation between the upregulated calcium/cation exchange protein genes (*OsCCX2* and *CCX5_Ath*) and ABC transporter gene *AB2E_Ath* with multiple upregulated DAMs in the salt-tolerant rice variety. Similarly, in the interaction network of differentially expressed transcription factors and 19 upregulated metabolites in TLJIAN, 6 NACs, 13 AP2/ERFs, and the upregulated WRKY transcription factors were positively correlated with 3 flavonoids, 3 lignans, and the lipid oleamide. These results suggested that the combined effects of differentially expressed transcription factors, transporter genes, and DAMs contribute to the enhancement of salt tolerance in TLJIAN. Moreover, this study provides a valuable gene–metabolite network reference for understanding the salt tolerance mechanism in rice.

## 1. Introduction

Soil salinization is one of the most significant abiotic stresses that affects various crop growth, development, and yield worldwide [1]. Except for in coastal areas, improper irrigation, the use of poor-quality water, and climate change particularly aggravate the salinization of arid and semi-arid soil in inland areas, which generally impairs the ability of the root to absorb water, leading to the disruption of metabolic processes and reduced photosynthetic efficiency [2,3]. Under salinity conditions, osmotic, ionic, and oxidative stresses affect most plants, inducing morphological and biochemical changes [4]. Plants respond to salt stress through many strategies, including the selective preferential uptake and exclusion of K^+^ ions over Na^+^ ions [5] and the elimination of reactive oxygen species (ROS) through the antioxidant defense system [6]. Rice (*Oryza sativa* L.) is a crucial crop globally, serving as the staple food for billions of people, especially those in Asian countries [7,8]. However, rice plants are inherently sensitive to salt stress, which is particularly severe in upland rice due to its cultivation in rain-fed croplands [7,8]. Moreover, rice is more sensitive during the early seedling and reproductive stages [9]. Therefore, it is crucial to explore new rice germplasms with salt tolerance and understand the underlying molecular mechanisms in order to breed salt-tolerant rice and ensure global food security [10].

In recent decades, there has been significant progress in the functional validation of a multitude of salt-responsive genes in rice, spanning from the sensing of salt to the transcriptional regulation of key genes [11]. At the transcriptomic level, plant responses to salt are known to involve large numbers of genes functioning in the stress signaling, transcription regulation, ion transport, and biosynthesis of specific metabolites of complex signaling pathways [12]. Transcription factors (TFs), which regulate gene expression by binding to the cis-acting elements in the promoter, play important roles in plant growth, development, and response to environmental stresses [13]. Through RNA-seq-based transcriptome analysis, it has been discovered that various TF families, such as NAC, MYB, WRKY, AP2/ERF, bZIP, and bHLHs, play a crucial role in plants responses to salinity stress [14]. The loss of *OsWRKY54* resulted in greater Na accumulation in shoots and enhanced sensitivity of rice plants to salt stress [15]. In wheat (*Triticum aestivum*), the expression of *TaWRKY75-A* was strongly induced by salt and polyethylene glycol (PEG) treatments. The heterologous expression of *TaWRKY75-A* in Arabidopsis showed the increased resistance of the plants to salt and drought stresses in transgenic plants, indicating that *TaWRKY75-A* might be involved in salt and drought resistance in wheat [16]. ONAC106 functions in salt tolerance, as well as leaf senescence and tiller angle determination, by modulating the expression of target genes that are involved in each specific signaling pathway [17].

Metabolites are the final output of plant responses to various environmental stresses [10]. Metabolomics-assisted breeding is of great significance in the holistic understanding of an organism from genome to phenome [18]. Recent reports have highlighted the significance of metabolites associated with salt stress tolerance in crop plants [19]. For example, cotton adapted to osmotic stress under salt stress by accumulating metabolites such as amino acids, sugars, and organic acids [20]. Soil salinity stress enhanced the accumulation of metabolites such as amino acids, alkaloids, organic acids, and polyphenols in barbary wolfberry fruits [21]. *Dendrobium officinale* had abundant metabolites of flavonoids, sugars, and alkaloids in leaves induced by salt stress [22]. Additionally, for salt-induced osmotic stress, a plant increases the accumulation of various osmo-protectants, such as proline, soluble sugar, betaine, trehalose, etc. [23].

Salinity stress responses in rice are highly complex and interconnected, involving a network of genes that coordinate salinity tolerance [24]. At the metabolome level, plants may exhibit a conserved metabolic response to salt stress, characterized by a change in the balance between amino acids and organic acids [25]. However, the specific responses of metabolites to salt stress vary depending on factors such as time, tissue, and cultivar [26]. Obviously, it is insufficient to solely examine the transcriptome or metabolome to unveil the regulatory mechanism of salt tolerance, therefore the integration of multi-omics data is necessary to gain a more comprehensive understanding of the genes and molecular signals associated with complex agronomic traits, such as stress tolerance. Despite advancements, the potential regulatory mechanism of salt tolerance, particularly the differences in salt stress tolerance among different genotypes, remains incompletely understood [27]. Recently, the integrative analysis of transcriptome coupled with metabolome has emerged as a widely used tool in studying molecular mechanisms and accelerating the development of salt-resistant and -tolerant rice varieties [9].

In this study, the differences in response to salt stress between different rice varieties were resolved by the conjoint analysis of the differential expression genes (DEGs) identified from the transcriptome and the differentially accumulated metabolites (DAMs) identified from the metabolome. The significantly upregulated transporter genes (*OsCCX2* and *CCX5_Ath*, *AAP3_Ath*, *AB2E_Ath* and *OCT7_Ath*) in salt-tolerant rice TLJIAN were positively correlated with multiple upregulated differential metabolites in this salt-tolerant rice. Furthermore, the upregulated transcription factors, such as NAC family, AP2/ERF family, and WRKY, in the TLJIAN were also positively correlated with the coordinately upregulated three flavonoid metabolites (dihydrokaempferide, isovitexin-8-O-xyloside, kaempferol -3-O-arabinoside-7-O-rhamnoside), three lignans and coumarins metabolites (matairesinol, matairesinoside, and lariciresinol-4′-O-glucoside), as well as one lipid metabolite (Oleamide) in this salt-tolerant variety. These findings will provide technical support for the exploration of the potential mechanism of salt tolerance and the selection of salt-tolerant rice varieties.

## 2. Results

### 2.1. Transcriptome Differences of Rice Varieties with Different Salinity Tolerance

To investigate the disparity in salt tolerance between two types of rice, namely salt-tolerant Lianjian 5 (TLJIAN) and salt-sensitive Huajing 5 (HJING), a comparative analysis of the transcriptomes was conducted. A total of 1518 differentially expressed genes (DEGs) were identified in the two types of rice cultivated in saline–alkali land, of which 882 genes were upregulated and 636 genes were downregulated in salt-tolerant rice (Appendix A). Among the DEGs, 935 of them had clear annotation information, including 66 transcription factor TFs, 33 transporters, and 61 receptor-like-kinase. Interestingly, 46 of these genes have been previously reported to be associated with salt-tolerance (Appendix A).

To gain a deeper understanding of the resulting DEGs functionality, MapMan software was used to locate these DEGs to the modules and biochemical or molecular biology pathways of the Arabidopsis Seed-Molecular Networks blueprint. In general, almost all differentially expressed transcription factors, such as NAC, WRKY, MYB, and AP2/ERE family, were upregulated in salt-tolerant rice, and *OsbHLH6*, *OsWRKY22*, *ERF17_Ath*, *OsDREB1B*, *OsDREB1C*, *OsDREB1F*, and *OsDREB1G* were significantly upregulated among the differentially expressed transcription factors in the major classes. The genes of stress-related DEGs, such as cold, drought/salt, and heat, were also upregulated in salt-tolerant rice, and the precursor synthesis genes of cell wall modules were all upregulated in salt-tolerant rice (Figure 1, Table 1). Both receptor-like protein kinase FERONIA (Os04g0586500, Os01g0769700) were upregulated in salt-tolerant rice (Figure 1). In addition, differentially expressed transporter genes *AB2E_Ath*, *OCT7_Ath*, *HIP7_Ath*, *NRAM6*, and *AB3C_Ath* were significantly upregulated in salt-tolerant rice (Table 2).

Compared to salt-sensitive differentially expressed genes, all three small molecules of heat shock protein were upregulated in salt-tolerant rice; all Early Responsive to Dehydration, Scarecrow-like Protein, and Accelerated Cell Death were also upregulated in salt-tolerant rice, but photosynthesis, fermentation genes and five of the seven genes involved in cell wall modification were downregulated in salt-tolerant rice. Peroxidase, UDP-glycosyltransferase, and Xylanase inhibitor were all raised, whereas 8 of the 9 genes involved in auxin synthesis and signaling were downregulated, as was the case for gibberellin, brassinosteroid, and jasmonate (Figure 1).

To gain insight into the biological significance of DEGs, GO and KEGG enrichment analyses were performed. GO enrichment analysis showed that DEGs were enriched in response to stress, maintaining the integrity of cell anatomy, catalytic activity, and transcriptional regulatory activity (Figure 2A), as well as enriched in response to environmental signal response processes, especially those related to immune responses (Figure 2B). KEGG pathway enrichment analysis further confirmed that DEGs were mainly enriched in anabolic pathways such as environmental signaling response plant–pathogen interaction, plant hormone signaling, MAPK signaling pathway, and amino sugar and nucleotide sugar metabolism (Figure 2C).

### 2.2. Metabolome Differences of Rice Varieties with Different Salt Tolerance

Plants required the production of numerous metabolites during growth and development to adapt to environmental changes and withstand various stresses [46]. To examine the metabolism differences between TLJIAN and HJING rice varieties under salt stress, UPLC-MS/MS was used to perform metabolomic analysis. A total of 1336 metabolites were detected in both types of rice, with 42 metabolites showing upregulation (|log2FC| ≥ 1, VIP > 1) in the salt-tolerant TLJIAN 5 compared with salt-sensitive HJING. Notably, the flavonoids dihydrokaempferide and isovitexin-8-O-xyloside, the lignans matairesinol and matairesinoside, the amino acid derivative N-Acetyl-L-Tryptophan, and the alkaloids octadecadienoic acid amide were significantly upregulated in the salt-tolerant rice. In addition, pyrocatechol, catechin, trehalose 6-phosphate, naringin, quercetin-5-O-β-D-glucoside, and lysophosphatidylcholine (20:5 and 16:4) were also found to be upregulated in the salt-tolerant rice, whereas the majority of organic acids were downregulated in salt-tolerant rice (Table 3).

### 2.3. Conjoint Analysis of Transcriptomic Difference and Metabolomic Fluctuation Accompanied with Different Salt Tolerance in Rice

In order to understand the difference mechanism of salt stress response between the two types of rice, a conjoint analysis of its transcriptome and metabolome data was performed in this study. The differentially expressed genes (DEGs) and differentially accumulated metabolites (DAMs) were mapped onto the KEGG pathway map to identify the pathways enriched with DEGs and DAMs. The major pathway with the highest enrichment was found to be amino acid biosynthesis, starch and sucrose metabolism, and phenylpropane biosynthesis pathways (Figure 3). Among them, amino acid biosynthesis was the pathway with the highest degree of differential enrichment of the KEGG pathway, including 11 DAMs and DEGs, respectively. It could be seen that, during the process of salt stress, the process of amino acid biosynthesis had the greatest influence on the root system of rice, and played an important role in regulating the process of salt tolerance stress in rice.

To further study the interaction between transcriptome and metabolome more systematically, gene–metabolite combinations with coefficient correlation (PCC > 0.80 and *p*-value < 0.05) were screened, and then stress-tolerance-related genes and metabolites were selected to obtain 29 differentially expressed transporter genes (Appendix A), 63 differentially expressed transcription factors, and 19 differentially expressed upregulated metabolites (Appendix A). Thus, transporter gene–metabolite and transcription-factor–metabolite-related network maps were constructed to show the relationship between salt-tolerance-related transcription factors and transporter genes and differential metabolite accumulation in the transcriptome under salt stress.

For the interaction network of transporters and upregulated metabolites, the calcium/cation exchange protein *OsCCX2* and *CCX5_Ath*, amino acid transporter *AAP3_Ath*, ABC transporter *AB2E_Ath*, and organic cation/carnitine transporter *OCT7_Ath* were positively correlated with multiple upregulated differential metabolites, such as dihydrokaempferide, isovitexin-8-O-xyloside, kaempferol-3-O-arabinoside-7-O-rhamnoside, catechin, and matairesinol, matairesinoside and magnolignan A-2-O-glucoside, etc. (Figure 4). For the interaction network, plots of differentially expressed transcription factors and 19 upregulated metabolites, 6 members of the NAC family and 13 members of the AP2/ERF family, and upregulated WRKY transcription factors were positively correlated with 3 flavonoid metabolites (dihydrokaempferide, isovitexin-8-O-xyloside, and kaempferol -3-O-arabinoside-7-O-rhamnoside) and 3 lignans and coumarins (matairesinol, matairesinoside, and lariciresinol-4′-O-glucoside) (Figure 5). In addition, catechin was positively correlated with *OsERF096*, *ERF12_Ath*, *ERF78_Ath*, *ERF92_Ath*, and *OsWRKY7*. Pyrocatechol was positively correlated with *OsDREB1H*, *OsWRKY24*, and *OsWRKY15*. *OsWRKY7* was positively correlated with quercetin glucoside (Figure 5).

## 3. Discussion

Under environmental stress, plants have evolved a variety of protective mechanisms, such as changing the content of related metabolites and regulating the expression of DEGs to resist stress damage, but a comprehensive understanding of their complex protection mechanisms is still an increasingly urgent need [47,48]. Currently, a range of omics techniques have emerged as effective strategies for systematically revealing the stress response mechanism [46]. Genes responsive to salinity have been categorized based on their functions, including transcription factors (TF), transporters, carbohydrates, protein metabolism, energy metabolism, hormones, reactive oxygen network components, osmoprotectants, cell walls, and signal transduction components [49]. In this study, a comparative transcriptome comparative analysis of different salt-tolerant rice varieties showed that 1518 DEGs of salt-tolerant TLJIAN responded to salt stress compared to salt-sensitive HJING, and these DEGs were functionally categorized into the aforementioned salt-responsive clusters (Figure 1, Table 2 and Appendix A).

The intensive studies indicated that various TFs are involved in improving salinity tolerance in rice [24]. Major Transcription factors (TFs) families regulated salt tolerance in rice such as dehydration-responsive element binding protein (DREB), ABA-responsive element binding protein/factor, and NAC [9]. The expression of the MYB, bHLH, NAC, ERF, WRKY, and bZIP transcription factors was altered under salt stress [50]. Intriguingly, almost all differentially expressed transcription factors, such as the NAC, WRKY, MYB, and AP2/ERE family, were upregulated in salt-tolerant rice, and several of them were demonstrated to be positive controllers of salinity tolerance in previous studies (Figure 1, Table 2 and Appendix A). *ONAC022*, a stress-responsive transcriptional activator, improved drought and salt stress tolerance in rice through modulating an ABA-mediated pathway [32]. *OsNAC14* mediated drought tolerance by recruiting factors involved in DNA damage repair and defense response resulting in improved tolerance to drought [51]. The important regulator of ethylene signal transduction, a transcription factor of the AP2/ERF family, also played an important role in plant growth and development and responded to biotic and abiotic stresses, such as the salt-treatment-induced expression of five ERF genes (*SpERF*) in tomatoes [13]. In this study, the AP2/ERF family showed an enhanced expression of members (including *ERF17_Ath*, *OsDREB1B*, *OsDREB1C*, *OsDREB1F*, and *OsDREB1G*) in salt-tolerant rice (Table 1). *OsDREB1C*-overexpressing rice boosted grain yields, shortened the growth duration, and conferred higher nitrogen use efficiency and early flowering [52]. The overexpression of a rice *OsDREB1F* gene increased salt, drought, and low temperature tolerance in both Arabidopsis and rice through the ABA-dependent pathway [34].

Besides these positive regulators, certain genes act as the negative controllers of salinity tolerance in rice [24]. Notably, salt-tolerant rice TLJIAN showed the downregulation of several known salt-tolerance-negative regulator genes, including *Drought and Salt Tolerance Coactivator 1* (*DCA1*) (Appendix A), a negative regulator of salt stress, which is consistent with the report where the downregulation of *DCA1* enhanced drought and salt tolerance [42]. Incremental studies suggest that numerous TFs and their interacting proteins have been implicated in rice salt stress response via regulating a series of signaling pathways [11]. Therefore, further exploration of these regulons, their interacting proteins, and target genes will contribute to understanding the network of signaling pathways involved in salt stress response and accelerate the elucidation of salt-tolerance mechanisms [11].

In addition, compared with salt-sensitive rice, many upregulated transporter genes in response to salt stress were found in salt-tolerant rice, among which two ABC transporter (*AB3C_Ath* and *AB2E_Ath*) genes were significantly upregulated. In plants, ABC transporters play an important role in stress tolerance [53]. Transgenic Arabidopsis plants that overexpress an ATP-binding cassette (ABC) transporter, *AtABCG36*/*AtPDR8*, were more resistant to drought and salt stress and grew to higher shoot fresh weight (FW) than the wild-type [54]. The expression of the *ABCG* gene in xylem, leaf, and apical buds was upregulated under NaCl stress in Betula halophila [53]. Therefore, the investigation of whether these transporters is regulated by salt-tolerant rice enhanced TFs (e.g., *NACs*, *OsDREB1* subfamily) under salt stress is of particular interest and warrants further research. Regarding the receptor-like protein kinase, two FERONIA (FER) members (Os04g0586500, Os01g0769700) were coordinately upregulated in salt-tolerant rice (Figure 1). FERONIA coordinated plant growth and salt tolerance via the phosphorylation of phytochrome B [55]. Under low temperature stress conditions, the FERONIA receptor kinase MdMRLK2 could enhance the cold tolerance of apple plants by increasing the accumulation of osmoregulatory substances in cells, the synthesis of anthocyanins, and maintaining the stability of cell wall components [56].

Some recent studies in metabolomics have reported that salt stress induced the accumulation of some compounds in *D. officinale* leaves, especially flavonoids, sugars, and alkaloids, which may play an important role in the salt-stress responses of leaf tissues from *D. officinale* [22]. Soil salt–alkaline stress enhanced the metabolites accumulation of amino acids, alkaloids, organic acids, and polyphenols contents in wolfberry fruits [21]. Furthermore, Wei et al. demonstrated that the overexpression of the phenylcoumaran benzylic ether reductase gene (*PtPCBER*) in the phenylpropane metabolic pathway improved the salt tolerance of poplar through lignan-mediated reactive oxygen species clearance [57]. Studies of mungbean reported that the contents of phenylpropanoid-derived metabolites, including catechin, chlorogenic acid, isovitexin, p-coumaric acid, syringic acid, ferulic acid, and vitexin, significantly increased under salinity conditions [58]. The salt stress study of Arabidopsis found that *MYB111* was a positive regulator in salt stress response, and that the additions of bioflavonoids and chalcone/dihydrokaempferole/quercetin were able to rescue the loss of salt tolerance in *myb111* mutants, which proved that flavonoids were crucial against salt stress [59].

Sugars are energy transporters and essential hydrophilic solutes that protect cellular proteins and membranes from extreme environmental challenges. Trehalose, an important osmolyte and osmo-protectant, and its intermediate compounds effectively modulate salt response and salt tolerance in different plants [60]. The variations in trehalose and trehalose-6-phosphate concentrations impact many biological functions, including adaptive stress responses [61]. In this study, differential metabolites in the flavonoids dihydrokaempferide, isovitexin-8-O-xyloside and catechin, and lignans matairesinol and matairesinoside, as well as trehalose-6-phosphate, were up-regulated in salt-tolerant rice (TLJIAN); meanwhile, the majority of organic acids were downregulated in salt-tolerant rice (Table 3). Under stress circumstances, the reduction in organic acid intermediates may be interpreted as a general consequence of decreased de novo CO_2_ assimilation of carbon dioxide under stress because of stomatal limitation [25]. When compared to the salt-sensitive rice IR64, the salt-tolerant variety FL478 displayed more significant increases in amino acids and sugars, but more noticeable decreases in organic acids in both leaves and roots [62]. In *Lotus japonicus*, the simultaneous decrease in the concentration of most organic acids may serve multiple purposes in maintaining vitality under salt stress [25]. Therefore, the function of fluctuated metabolites between TLJIAN and HJING identified here in rice salt tolerance deserves in-depth study.

To better understand the relationship between differential genes and differential metabolites and their role in salt tolerance, combined transcriptome and metabolome analysis may be the best approach [63]. The combined transcriptome and LC-ESI-MS/MS metabolomics analysis revealed that melatonin enhanced plant tolerance to multiple stresses by mediating flavonoid biosynthesis, providing new ideas for studying the crosstalk between melatonin and flavonoids [64]. Previous studies have reported that trehalose-6-phosphate synthase (TPS) plays a crucial role in the biosynthesis of trehalose. Furthermore, rice *OsTPS1* and *OsTPS8* have been found to positively regulate salt tolerance by enhancing the accumulation of trehalose and proline in rice overexpression plants [65,66]. In this study, an up-regulation of trehalose-6-phosphate was observed in salt-tolerant TLJIAN (Table 3), concomitant with the increased expression of trehalose-6-phosphate synthase *OsTPS1* (Appendix A).

Transcriptome and metabolomic analysis found that genes involved in the synthesis pathways of phenylpropanin, ferulic acid (FA), and spermidine (Spd) in tomatoes, especially transcription factors such as MYB, Dof, BPC, and AP2/ERF, were expressed under salt stress [67]. Combined transcriptome and metabolome analysis under watermelon salt stress showed that genes involved in phenylpropanin synthesis, hormone signaling, and carbohydrate synthesis pathways, as well as bHLH family transcription factors, played an important role in improving the salt tolerance of watermelon [68]. In this study, the metabolites and genes with significant differences were analyzed for correlation, and by constructing a network map of transporters/transcription factors–metabolites. It was found that transporters, including cation/calcium exchange proteins *OsCCX2* and *CCX5_Ath*, amino acid transporter *AAP3_Ath*, ABC transporter *AB2E_Ath*, and organic cation/carnitine transporter *OCT7_Ath*, were significantly upregulated under salt stress (Figure 4, Table 2). The expression of transcription factors of 6 NAC family members, 13 AP2/ERF family members, and 9 WRKY family members was significantly upregulated under salt stress and positively correlated with 3 flavonoids (dihydrokaempferide, isovitexin-8-O-xyloside and kaempferol-3-O-arabinoside-7-O-rhamnoside), 3 lignans (matairesinol, matairesinoside and Lariciresinol-4′-O-glucoside), and 1 lipid metabolite (oleamide) (Figure 5, Table 1). These results suggested that the transporters and transcription factors helped to improve the salt tolerance of TLJIAN rice.

## 4. Materials and Methods

### 4.1. Plant Materials and Salt Treatment

Two japonica rice (*O. sativassp*. japonica), including the salt-tolerant rice variety Lianjian 5 (TLJIAN) and salt-sensitive variety Huajing 5 (HJING), were cultivated in the Qingkou salt field experimental base (salt content in soil > 0.5%, electrical conductivity EC ~2.32 S m^−1^) at Jiangsu Lianyungang Academy of Agricultural Sciences in 2022. Rice seeds were sown on May 15 and transplanted on June 15. During the growth period, except the tillering-hearty period, the water layer was maintained in the field using the irrigation water (salt content ~0.15%). The roots were sampled on August 28 (7 days after the heading stage), placed in sterilized 5 mL centrifuge tubes, and stored in a −80 °C refrigerator before extracting RNA, and three biological replicates were set up per sample.

### 4.2. RNA Extraction and Transcriptome Sequencing

First, TRIzol reagent was used to extract high-quality RNA (no degradation and no DNA contamination) from each sample, then a NanoPhotometer spectrophotometer was used to measure RNA purity (OD 260/280 and OD 260/230 ratios), then the RNA concentration was accurately quantified using a Qubit2.0 Fluorometer, and finally Agilent 2100 was used as a Bioanalyzer to accurately detect RNA integrity. Illumina’s NEBNext^®^ kit was used to build the library with total RNA > 1 ug as the starting RNA. We enriched mRNA with polyA tails using Oligo(dT) magnetic beads. A fragmentation buffer was then added to break the RNA into short fragments, and double-stranded cDNA was synthesized using the short RNA as a template, with three biological replicates. After pooling different libraries according to the effective concentration and 5 Gb data volume, high-throughput sequencing was performed using the Illumina NovaSeq 6000 platform to generate 150 bp paired end sequencing data.

### 4.3. Bioinformatics Analysis of Transcriptome

We used fastp (v0.19.3) to filter raw data, removed adapters and low-quality reads, and obtained clean reads for subsequent analysis [69]. The reference genome was the genome sequence IRGSP-1.0_genome.fasta.gz (https://rapdb.dna.affrc.go.jp/download/irgsp1.html, 1 November 2022) of Japonica Nipponbare, indexed using HISAT2 [70], and clean reads were compared to the reference genome [71].

The sequence of the new gene was extracted from the genome, and the new gene was associated with KEGG, GO, NR, using diamond software. Swiss-Prot, trEMBL, KOG database sequence alignment results were annotated with Evalue le-5 [72]. Plant transcription factor prediction used iTAK software, which integrated two databases, PnTFDB and PlantTFDB [73]. The gene alignment was calculated using featureCounts (v1.6.2), and then the FPKM of each gene was calculated based on the gene length. Gene differential expression analysis between the two groups was performed using DESeq2 [74]. The adjusted *p* values < 0.05 and |log2foldchange| ≥ 1 as a threshold for identifying differentially expressed genes [74].

To analyze the pathways and functions of DEGs related to salt stress, MapMan (v3.6.0 RC1) was utilized to annotate and visualize DEGs on metabolic pathways [75,76].

### 4.4. Enrichment Analysis of Differentially Expressed Genes

Enrichment analysis was performed based on hypergeometric testing, and a hypergeometric distribution test of KEGG was performed on pathway units and genetic annotation was performed [77]. Using GO-Term in the GO database, a hypergeometric test of GO was performed to find GO-Term, which was significantly enriched in differentially expressed genes compared to the entire genome background [78]. Diamond software was used to align protein sequences or cDNA sequences to the KOG database, and then extract the annotations of the KOG database [79].

### 4.5. Sample Preparation and Extraction for Widely Targeted Metabolomics

The three biological samples from TLJIAN and HJING, respectively, were freeze-dried with a vacuum freeze dryer (Scientz-100F), then the freeze-dried samples were crushed using a mixer with zirconia beads (MM 400, Retsch) at 30 Hz for 1.5 min. Next, 50 mg of lyophilized powder was dissolved in 1.2 mL of 70% methanol solution and vortexed for 30 s every 30 min for a total of 6 times. After centrifugation at 12,000 rpm for 3 min, the extract was filtered prior to UPLC-MS/MS analysis.

### 4.6. Metabolomic Analysis

For the quantitative analysis of metabolites, we employed a multiple reaction monitoring (MRM) method using the triple-quadrupole mass spectrometry. QQQ scanning used MRM mode with collision gas (nitrogen) set to medium. Through further DP (decomposition potential) and CE (collision energy) optimization, the DP and CE of each MRM ion pair are obtained. A specific set of MRM ion pairs was monitored during each period based on the metabolites eluted in each epoch [79].

Based on the NIST compound mass spectrometry library and the metabolome database of the Wuhan Metware Company (www.metware.cn, 1 November 2022), the mass spectrometry data were processed using software Analyst 1.6.3 to qualitatively quantify the metabolites of the samples. After the qualitative and quantitative analysis of the detected metabolites, combined with the grouping of specific samples, on the one hand, based on the OPLS-DA results, from the variable importance projection (VIP) of the obtained multivariate analysis OPLS-DA model, the metabolites with differences between different materials were preliminarily screened. On the other hand, the fold change (FC) of metabolites in each group was compared to further screen for differential metabolites. Differentially accumulated metabolites (DAMs) between two rice varieties were determined using VIP ≥ 1 and absolute Log2FC (fold change) ≥ 1 [10,63].

### 4.7. Combined Analysis of Transcriptome and Metabolome

To better understand the interactions between transcriptome and metabolome in rice roots under salt stress, for each differentially accumulated metabolite (DAM) (|log2FC| ≥ 1, VIP > 1), all DEGs were submitted to analyze metabolite–gene correlation according to the Pearson correlation coefficients (PCCs) [63].

DEGs and DAMs were mapped simultaneously to the KEGG pathway database, further according to the KEGG enrichment analysis results of DEGs and DAMs, the KEGG metabolic pathway that enriched the DEGs and DAMs at the same time was obtained. Then, the functional analysis and the correlation analysis of DEGs and DAMs were combined to screen key metabolic pathways, genes, and metabolites [10].

## 5. Conclusions

The combined analysis of transcriptome and metabolome in this study provides insights into the molecular mechanisms of salt tolerance in rice landrace TLJIAN. The results suggest that the molecular mechanisms of salt tolerance in rice plants might involve amino acid biosynthesis, starch and sucrose metabolism, and phenylpropane biosynthesis pathways. Additionally, the interaction network analysis reveals that the significantly upregulated transporter genes *OsCCX2*, *CCX5_Ath*, *AAP3_Ath*, *AB2E_Ath*, and *OCT7_Ath* in TLJIAN were positively correlated with multiple upregulated differential metabolites. The upregulated transcription factors, such as NAC family, AP2/ERF family, and WRKY, in salt-tolerant rice were positively correlated with specific flavonoid metabolites, lignans and coumarins metabolites, and lipid metabolites, suggesting that the differentially expressed transcription factors, transporter genes, and metabolites played a crucial role in the resistance of salt-tolerant varieties to salt stress. Overall, this study highlights the differences in transcriptome and metabolome between rice with different salinity tolerances, and provides a gene–metabolite network reference for revealing the mechanism of rice salt tolerance.

## Figures and Tables

**Figure 1 plants-12-03359-f001:**
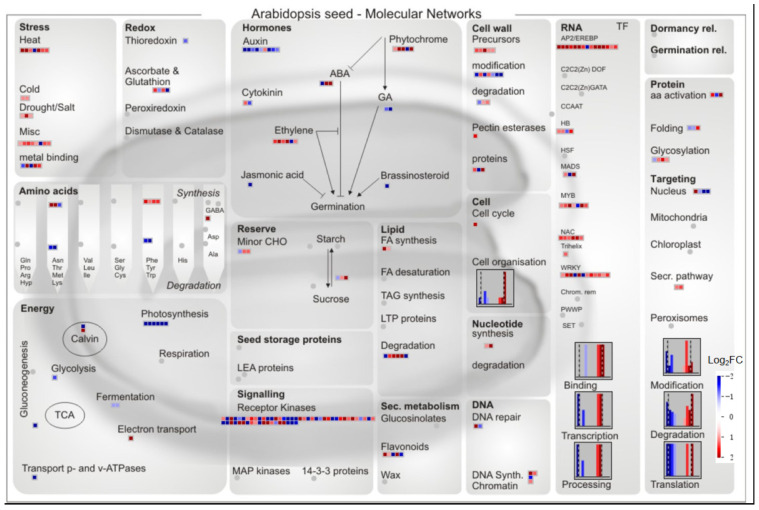
**Mapman visualization of differentially expressed genes in two rice varieties with different salt tolerance.** Red blocks indicate the upregulated genes and blue blocks indicate the downregulated genes. Log_2_FC means log_2_foldchange (TLJIAN/HJING).

**Figure 2 plants-12-03359-f002:**
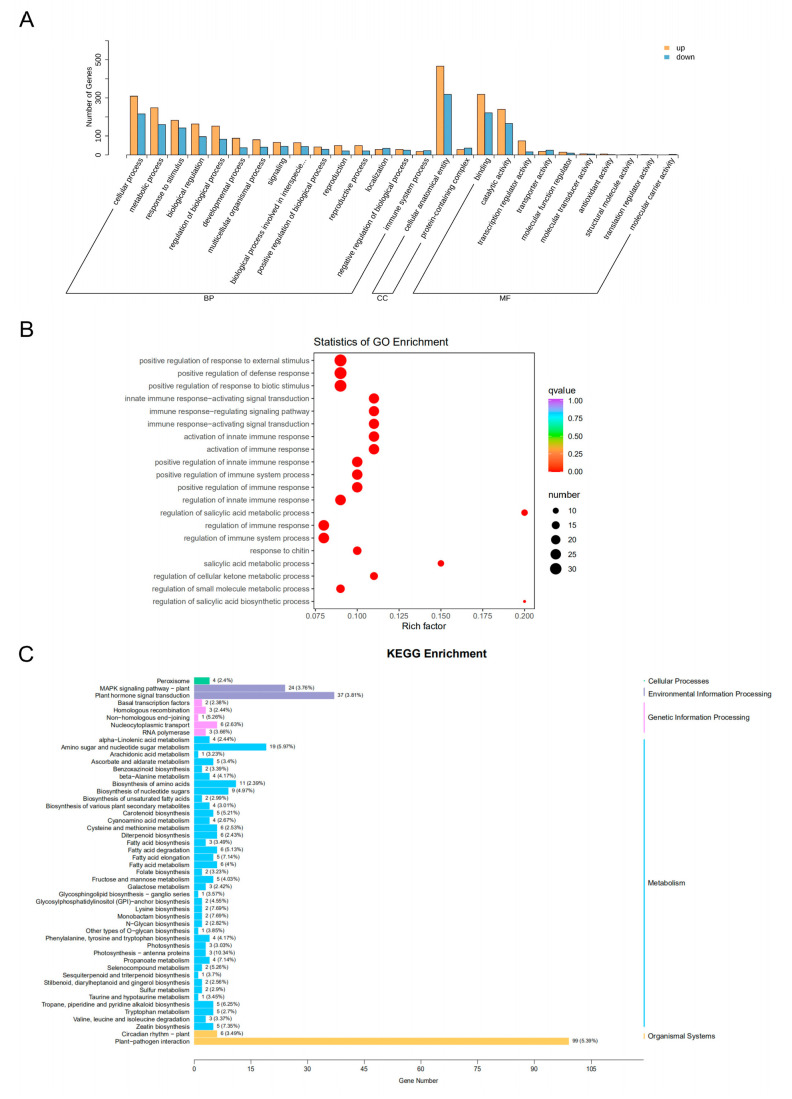
**GO and KEGG enrichment analysis of differentially expressed genes in two rice varieties with different salt tolerance.** (**A**) GO classification chart. BP: biological process; CC: cellular component; MF: molecular function. (**B**) GO bubble chart. (**C**) KEGG classification.

**Figure 3 plants-12-03359-f003:**
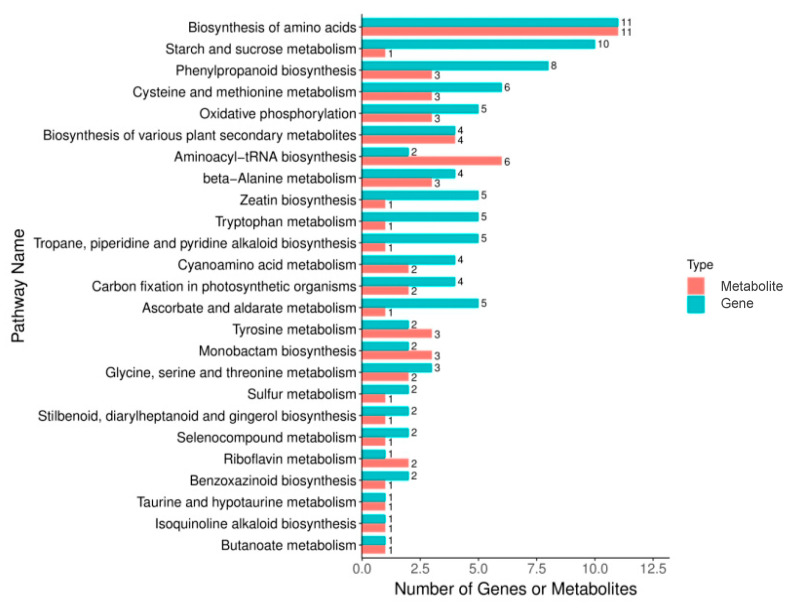
The bar plot of differentially expressed genes and differentially accumulated metabolites post KEGG enrichment analysis.

**Figure 4 plants-12-03359-f004:**
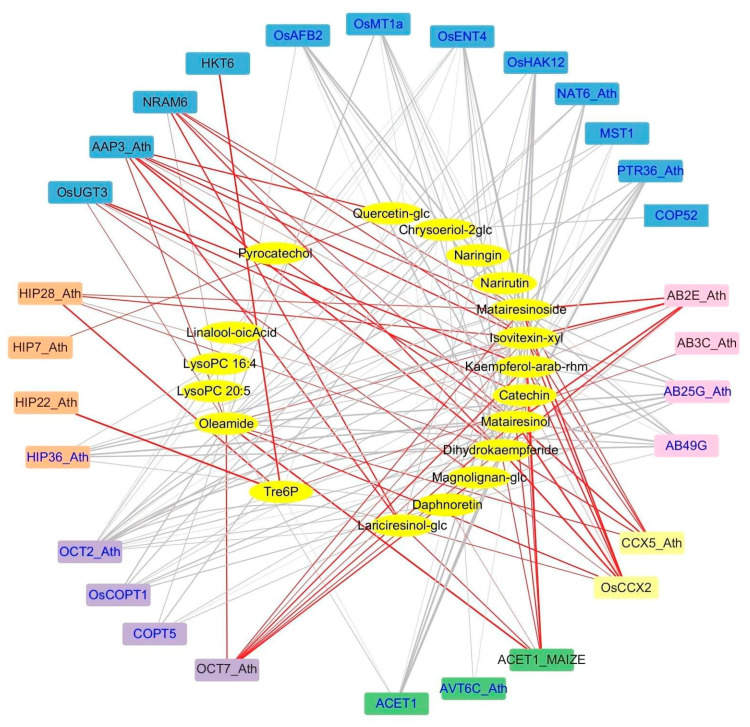
**Correlation between differentially expressed transporter genes and differentially accumulated metabolites in TLJIAN and HJING.** Rectangles represent transporter genes; ovals represent metabolites; red and gray lines represent positive and negative correlations, respectively; the thickness of the lines indicates the strength of the correlation, and the thicker the line, the stronger the correlation.

**Figure 5 plants-12-03359-f005:**
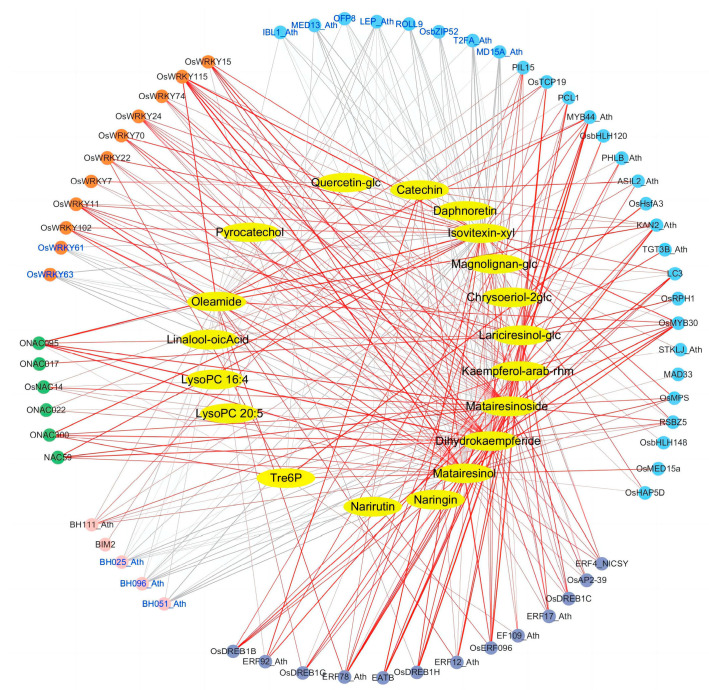
**Correlation between differentially expressed transcription factors and differentially accumulated metabolites in TLJIAN and HJING.** Circles represent transcription factors while ovals represent metabolites; red and gray lines represent positive and negative correlations, respectively; the thickness of the lines indicates the strength of the correlation, and the thicker the line, the stronger the correlation.

**Table 1 plants-12-03359-t001:** Transcription factors differentially expressed in TLJIAN and HJING.

Locus	Name	Log_2_FC	*p*_Value	Locus	Name	Log_2_FC	*p*_Value
**bZIP**				Os01g0821600	*OsWRKY21*	1.8	3.4 × 10^−2^
Os03g0127500	*OsbZIP25*	1.9	7.6 × 10^−3^	**Os05g0537100** [28]	** *OsWRKY7* **	1.1	3.9 × 10^−2^
novel.539	*RSBZ5*	1.1	5.1 × 10^−3^	Os09g0334500	*OsWRKY74*	1.6	5.2 × 10^−3^
**Os06g0662200** [29]	*OsbZIP52*	−1.2	1.8 × 10^−2^	Os01g0656400	*OsWRKY15*	1.5	2.0 × 10^−3^
**Heat-shock TF**				Os11g0686250	*OsWRKY63*	−9.4	3.0 × 10^−12^
Os02g0527300	*OsHsfA3*	1.3	1.5 × 10^−2^	**Os11g0685700** [28]	** *OsWRKY61* **	−10	1.9 × 10^−13^
**bHLH**				**AP2/ERF**			
Os02g0671300	*BH111_Ath*	1.2	4.1 × 10^−4^	Os09g0572000	*ERF92_Ath*	1.5	9.8 × 10^−3^
**Os03g0741100** [30]	** *OsbHLH148* **	1.7	4.0 × 10^−2^	Os04g0610400	*OsAP2-39*	1.4	1.8 × 10^−3^
Os12g0632600	*BH025_Ath*	−2	1.2 × 10^−8^	Os02g0157950	*ERF78_Ath*	1.4	5.0 × 10^−7^
**Os01g0159800** [31]	** *BH051_Ath* **	−2.1	2.2 × 10^−2^	Os05g0497300	*ERF4_NICSY*	1.2	7.6 × 10^−5^
Os04g0301500	*OsbHLH6*	2.2	7.3 × 10^−4^	Os10g0562900	*OsERF096*	1.9	4.7 × 10^−3^
Os08g0483900	*BH096_Ath*	−1.5	2.0 × 10^−2^	Os02g0767800	*ERF12_Ath*	1.8	2.7 × 10^−3^
Os08g0490000	*BIM2*	1.5	3.2 × 10^−2^	Os02g0781300	*ERF17_Ath*	2.8	1.9 × 10^−8^
Os09g0455300	*OsbHLH120*	1.3	2.7 × 10^−3^	Os02g0676800	*ERF25_Ath*	2	2.7 × 10^−2^
Os01g0286100	*PIL15*	1.4	5.0 × 10^−3^	Os05g0572000	*OsRPH1*	1.5	4.4 × 10^−2^
**NAC**				Os09g0457900	*EATB*	2	4.0 × 10^−4^
Os07g0566500	*ONAC010*	1.9	2.4× 10^−2^	Os08g0474000	*EF109_Ath*	1.7	3.0 × 10^−3^
**Os03g0133000** [32]	** *ONAC022* **	1.4	7.3 × 10^−3^	Os04g0399800	*LEP_Ath*	−1.9	7.1 × 10^−3^
Os06g0726300	*ONAC095*	1.3	2.9 × 10^−7^	**Os09g0522000** [30]	** *OsDREB1B* **	2.1	3.4 × 10^−3^
Os01g0675800	*OsNAC14*	1.4	2.0 × 10^−3^	**Os06g0127100** [33]	** *OsDREB1C* **	2.5	5.7 × 10^−5^
Os12g0123800	*ONAC300*	1.5	2.0 × 10^−6^	**Os01g0968800** [34]	** *OsDREB1F* **	2.5	1.5 × 10^−2^
Os11g0154500	*ONAC017*	1.9	1.2 × 10^−2^	**Os02g0677300** [35]	** *OsDREB1G* **	2.3	7.4 × 10^−3^
**Os01g0862800** [36]	** *ENAC1* **	1.3	1.1 × 10^−3^	Os09g0522100	*OsDREB1H*	1.7	1.5 × 10^−3^
**WRKY**				**Other**			
Os01g0626400	*OsWRKY11*	1.2	4.8 × 10^−3^	**Os02g0618400** [37]	** *OsMPS* **	1.4	2.7 × 10^−3^
**Os05g0474800** [38]	** *OsWRKY70* **	1.4	8.4 × 10^−4^	**Os03g0180800** [39]	** *OsJAZ9|OsTIFY11a* **	1.5	3.9 × 10^−2^
Os01g0826400	*OsWRKY24*	1.3	2.8 × 10^−3^	**Os01g0839100** [40]	** *ZFP179* **	1.3	4.7 × 10^−2^
Os01g0182700	*OsWRKY102*	1.1	4.7 × 10^−2^	**Os03g0820300** [41]	** *ZFP182|ZOS3-21* **	1.3	1.3 × 10^−3^
Os01g0820400	*OsWRKY22*	3.5	4.6 × 10^−4^	**Os10g0456800** [42]	** *DCA1* **	−1.7	1.5 × 10^−2^
**Os07g0460900** [28]	** *OsWRKY115* **	2	1.1 × 10^−7^				

Gene locus in bold means gene reported in previous studies, and the number in brackets represents the related reference. Log_2_FC means log_2_foldchange (TLJIAN/HJING).

**Table 2 plants-12-03359-t002:** Differentially expressed transporter genes in TLJIAN and HJING.

Locus	Name	Log_2_FC	*p*_Value	Annotation
Os09g0364800	*HIP22_Ath*	1.9	3.2 × 10^−3^	heavy-metal-associated domain-containing protein
Os01g0758000	*HIP28_Ath*	1.5	1.3 × 10^−3^	
Os07g0671400	*HIP36_Ath*	−2.8	4.8 × 10^−3^	
Os07g0298900	*HIP7_Ath*	3.3	4.2 × 10^−4^	
Os01g0733001	*NRAM6*	2.3	2.4 × 10^−6^	metal transporter Nramp
**Os11g0704500** [43]	** *OsMT1a* **	−1.2	2.4 × 10^−3^	Metallothionein-like
Os02g0288400	*AB3C_Ath*	2.4	2.3 × 10^−2^	ABC transporter
Os10g0432200	*AB2E_Ath*	9.9	4.4 × 10^−25^	
Os11g0177750	*AB25G_Ath*	−1.4	2.0 × 10^−4^	
Os12g0239900	*AB49G*	−1.4	6.6 × 10^−4^	
Os12g0181533	*AAP3_Ath*	2.6	8.2 × 10^−3^	amino acid transporter
Os12g0181600	*AAP3_Ath*	1.3	1.1 × 10^−5^	
Os01g0825800	*AVT6C_Ath*	−2.1	3.6 × 10^−3^	
Os08g0101700	*ACET1_Zma*	1.5	1.3 × 10^−2^	Ascorbate-specific transmembrane electron transporter
Os10g0504200	*ACET1*	−1.1	1.8 × 10^−3^	
Os10g0418100	*OsACA7*	1.9	3.0 × 10^−2^	calcium-transporting ATPase
Os01g0770700	*OsCOPT1*	−1.2	1.2 × 10^−3^	Copper transporter
Os05g0424700	*COPT5*	−1.9	1.8 × 10^−3^	
Os07g0557400	*OsENT4*	−1.6	1.4 × 10^−5^	Equilibrative nucleotide transporter 3
Os08g0420600	*NAT6_Ath*	−1.8	4.0 × 10^−2^	nucleobase–ascorbate transporter
Os07g0561800	*OCT2_Ath*	−1.9	2.2 × 10^−7^	Organic cation/carnitine transporter
novel.660	*OCT2_Ath*	−2.2	1.1 × 10^−4^	
Os03g0638200	*OCT7_Ath*	6.7	2.1 × 10^−5^	
Os02g0175000	*HKT6*	1.6	4.7 × 10^−2^	potassium transporter
Os08g0206400	*OsHAK12*	−1.5	1.7 × 10^−7^	potassium transporter
Os01g0268100	*PTR36_Ath*	−1.5	2.5 × 10^−3^	major facilitator
Os11g0104050	*SC61B_Ath*	1.2	3.0 × 10^−2^	protein transport protein
Os08g0455900	*COP52*	−1.9	3.3 × 10^−2^	copper transporter
Os02g0574000	*MST1*	−2.4	2.6 × 10^−2^	Sugar transport protein
**Os04g0395600** [44]	** *OsAFB2* **	−1.3	6.6 × 10^−3^	Transport inhibitor response TIR1-like
Os07g0581000	*OsUGT3*	1.2	5.6 × 10^−3^	UDP–galactose transporter
Os03g0656500	*OsCCX2*	1.5	1.0 × 10^−9^	Cation/calcium exchanger
**Os11g0148000** [45]	** *CCX5_Ath* **	1.2	1.3 × 10^−5^	sodium/calcium exchanger

Gene locus in bold means gene reported in previous studies, and the number in brackets represents the related reference. Log_2_FC means log_2_foldchange (TLJIAN/HJING).

**Table 3 plants-12-03359-t003:** Metabolites with significant differences in TLJIAN and HJING.

Compounds	CAS	Log_2_FC	VIP
**Phenolic acids**			
Methyl salicylate-2-O-glucoside	10019-60-0	1.48	1.82
Glucosyringic acid	-	1.46	1.83
Pyrocatechol	120-80-9	1.44	1.26
Rosmarinic acid	537-15-5	1.26	1.78
4-O-Sinapoylquinic acid	-	1.23	1.67
3-O-Methylgallic acid	3934-84-7	1.17	1.35
Methyl gallate	99-24-1	1.17	1.35
Methyl Cinnamate	103-26-4	1.16	1.41
Hydroxytyrosol	10597-60-1	1.10	1.78
1,2,2′-Trisinapoyl gentiobiose	-	1.09	1.29
Rosmarinic acid-3′-O-glucoside	910028-78-3	1.08	1.73
**Flavonoids**			
Naringin	10236-47-2	1.05	1.16
Narirutin	14259-46-2	1.05	1.16
Dihydrokaempferide	137225-59-3	12.76	1.94
Dihydromyricetin	27200-12-0	-1.05	1.38
Desmethylxanthohumol	115063-39-3	-1.64	1.35
Dihydrocharcone-4′-O-glucoside	-	-9.07	1.15
Phlorizin	60-81-1	-9.74	1.16
Isovitexin-8-O-xyloside	-	11.54	1.94
Chrysoeriol-8-C-glucoside-7-O-(6′’-feruloyl) glucoside	-	1.33	1.42
Kaempferol-3-O-arabinoside-7-O-rhamnoside	-	1.77	1.58
Quercetin-5-O-β-D-glucoside	-	1.11	1.01
Catechin	154-23-4	1.25	1.26
Epiafzelechin	24808-04-6	−1.19	1.16
Gallocatechin-(4α→8)-gallocatechin	-	−2.99	1.17
**Lignans and Coumarins**			
Matairesinol	580-72-3	5.75	1.92
Matairesinoside	23202-85-9	4.62	1.89
Magnolignan A-2-O-glucoside	1035846-18-4	1.27	1.50
Lariciresinol-4′-O-glucoside	143663-00-7	1.26	1.66
Daphnoretin	2034-69-7	1.79	1.56
Scopolin	531-44-2	−1.31	1.83
**Terpenoids**			
8-O-Acetylharpagide	6926-14-3	1.18	1.22
Catalpol	2415-24-9	−1.01	1.60
Dehydroabietic acid	1740-19-8	1.34	1.60
2,3-Dihydroxy-12-ursen-28-oic acid	-	−1.42	1.24
2,3,23-Trihydroxyolean-12-en-28-oic acid	-	−3.30	1.61
**Organic acids**			
2-Hydroxyhexadecanoic acid	764-67-0	1.31	1.32
4-Guanidinobutyric acid	463-00-3	1.25	1.72
Dibutyl phthalate	84-74-2	−1.00	1.11
Fumaric acid	110-17-8	−1.02	1.50
L-Malic acid	97-67-6	−1.04	1.47
3-Hydroxyglutaric acid	638-18-6	−1.19	1.35
3-Isopropylmalic acid	921-28-8	−1.40	1.43
2-Isopropylmalic acid	49601-06-1	−1.44	1.46
2-Propylmalic acid	-	−1.51	1.48
3-Ureidopropionic acid	462-88-4	−0.15	1.46
**Lipids**			
LysoPC 16:4	-	1.25	1.40
LysoPC 20:5	162440-04-2	1.10	1.27
LysoPC 16:2(2n isomer)	-	−1.07	1.56
LysoPE 17:0	-	−1.13	1.09
Oleamide	301-02-0	1.58	1.74
€-Linalool-1-oic acid	-	1.17	1.35
**Alkaloids**			
Dhurrin-6′-O-Glucoside	-	1.32	1.43
octadecadienoic acid amide	-	3.69	1.26
N-(4-Aminobutyl) benzamide	5692-23-9	1.03	1.78
2-Glucosyl-glucosyloxy-2-phenylacetic acid amide	-	1.06	1.83
**Amino acids and derivatives**			
N-Acetyl-L-phenylalanine	2018-61-3	1.25	1.79
N-Acetyl-L-Tryptophan	1218-34-4	3.32	1.91
3,4-Dihydroxy-L-phenylalanine (L-Dopa)	59-92-7	1.16	1.08
**Others**			
Trehalose 6-phosphate	4484-88-2	1.38	1.53
2-Aminoethylphosphonate	2041-14-7	1.09	1.29
D-Panthenol	81-13-0	1.24	1.72

CAS is the Chemical Abstracts Service Registry Number; Log_2_FC means log_2_foldchange (TLJIAN/HJING); VIP (Variable Importance in Projection) is an index used to evaluate the correlation between a certain metabolite and sample classification. The higher the VIP value, the greater the contribution of the metabolite to the sample classification.

## Data Availability

The data generated or analyzed during this study are available in the NCBI database repository (BioProject accession: PRJNA994662).

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
