# Peer review of "Integrated Analysis of Transcriptome and Metabolome Reveals Molecular Mechanisms of Rice with Different Salinity Tolerances"

_plants, 2023, doi:10.3390/plants12193359_

Round 1
Reviewer 1 Report
Please see the attached file for the comments.

Author Response
Response to Reviewer plants-2566860 Comments
Thank you for your valuable feedback and insightful suggestions on our manuscript. We have carefully revised the manuscript based on your recommendations. Below, we provide a point-by-point explanation addressing each of the comments.
Major Comments
- Introduction: Please explain the concept of how the integration of transcriptome and metabolome is performed and how this analysis can provide any insight information that can be advantage over other types of analysis.
Answer: Thanks for your suggestion on enhancing the research context of this manuscript. Regarding the integration of transcriptome and metabolome, for each differentially accumulated metabolites (DAMs) (|log2FC|≥1, VIP>1), all DEGs were submitted to analyze metabolite-transcript correlation according to the Pearson correlation method, and we have stated this in ‘Combined analysis of transcriptome and metabolome’ of Methods section.
Obviously, it is insufficient to solely examine the transcriptome or metabolome to unveil the regulatory mechanism of salt tolerance, therefore the integration of multi-omics data is necessary to gain a more comprehensive understanding of genes and molecular signals associated with complex agronomic traits, such as stress tolerance. we have stated this in the Introduction section.
- Materials and methods:
2.1 Explain how plants were grown. Are there any controls (no salt stress)? How many replications were performed?
Answer: The salt-tolerant Lianjian5 (TLJIAN) and salt-sensitive Huajing5 (HJING) were planted in the Qingkou salt field experimental based (salt content in soil > 0.5%) of Jiangsu Lianyungang Academy of Agricultural Sciences in 2022. Rice seeds were sown on May 15, transplanted on June 15. During the grown period except tillering-hearty-period, the water layer was maintained in field using the irrigation water (salt content ~0.15%). The roots were sampled on August 28 (7 days after heading stage). We have stated this in ‘Plant materials and salt treatment’ of Methods section.
2.2 Soil salinity level should be also indicated as soil EC (electrical conductivity).
Answer: The experimental salt field (salt content in soil > 0.5%, electrical conductivity EC ~2.32 S m-1)
2.3 Irrigated water should be indicated as EC or concentration of NaCl or ion concentration. Why blackish water was used in this experiment? Please clarify. Any deionized water was used in the control experiment? Please provide the information.
Answer: The concentration of NaCl in irrigated water is about 0.15%. No deionized water was used in the control experiment.
2.4 Please indicate the year of experiment. In the manuscript, only the dates were provided.
Answer: The experiment was performed in 2022, and the year has been stated in the manuscript.
2.5 It is quite unclear for the replication of RNA samples. Were they biological replications or technical replications?
Answer: Thanks for your concerns. In fact, three biological replicates were used in this study, which have been identified as biological replicates in the Materials and Methods section.
2.6 In Bioinformatics and RNA Seq data section, it was mentioned about new gene prediction. Are there any new genes predicted in this research? I could not find ones. If there are, please clarify clearly and explain how they (it) are related to salt stress response. If not, please revise the materials and methods section.
Answer: Yes, new genes mean without gene locus.
2.7 Add the reference for DESeq2. If your team created an analysis tool to identify differential expressed genes, please explain your method. What was the statistical method used to identify the differential expressed genes?
Answer: References for DESeq2 have been added. Genes with |log2FC|≥1 and P-values < 0.05 identified by DESeq2 were assigned as differentially expressed genes (DEGs).
2.8 What was the statistical analysis used to identify the difference in metabolites? Please explain.
Answer: Differential metabolites were screened based on |log2FC|≥1 and VIP>1, and screening metabolite criteria have been added in the main text.
2.9 The combined analysis of transcriptome and metabolome was unclear. It seems to be selected groups of genes were used for analysis with the metabolome. Is it a bias to do so? Please clarify.
Answer: For each differentially accumulated metabolites (DAMs) (|log2FC|≥1, VIP>1), all DEGs were submitted to analyze metabolite-transcript correlation according to the Pearson correlation method (|PCC|>0.8 and pvalue<0.05), then we can select groups of genes and metabolites for network analysis. At the same time, DEGs and DAMs were mapped simultaneously to the KEGG pathway database, further according to the KEGG enrichment analysis results of DEGs and DAMs, the KEGG metabolic pathway that enriched the DEGs and DAMs at the same time was obtained. Then, the functional analysis and the correlation analysis of DEGs and DAMs were combined to screen key metabolic pathways, genes and metabolites. We have stated this in ‘Combined analysis of transcriptome and metabolome’ of Methods section.
2.10 Please clarify the statistical analysis of how the edges linking between the genes and metabolites were generated. For correlation analysis, more than 1 timing of sample collection should be used. There was no information about timing of sample collection in this research. Please clarify.
Answer: In the joint analysis of genes and metabolites, the correlation is calculated between all samples of genes and all samples of metabolism. Then, a threshold screening is performed on the correlation results, considering a correlation coefficient (|PCC|) greater than 0.8 and a p-value less than 0.05. Subsequently, genes and metabolites of interest are selected based on their corresponding relationship within the correlation results. Finally, the correlation network diagram is created using cytoscape.
- Results:
3.1 In the transcriptome analysis, only the upregulated genes were mentioned. The salt stress response, both up-regulated and down-regulated are important depending on the functions of the genes. It will be better to focus on both up-regulated and down-regulated genes.
Answer: Thanks to the reviewer’s helpful suggestions. We have added the description of the downregulated genes to the Results and cited the related literatures in Tables and Discussion section.
3.2 Data in Table 1 and table 2 were not consistent with Figure 1 results.
Answer: Thanks for your concern. In fact, Figure1 displays those DEGs that can be mapped to the overview framework, while Table1 and Table2 are the transcription factors and transporter genes we have screened out, which are equivalent to part of the data in Figure1.
3.3 It is unclear when the up-regulated genes were mentioned. Does it mean the genes that were induced by salt stress condition or the genes that have higher expression in salt tolerant cultivar than salt sensitive cultivar? If the latter is the case, if we do not know the base of expression in both cultivars (no salt stress induction), this can lead to misinterpretation of the data. (Figure 1, 2, Table 1 and 2)
Answer: Thanks for the insightful comment. In fact, salt-tolerant and salt-sensitive rice cultivar were planted in salt field experiment base. The up-regulated genes mean that have higher expression in salt tolerant cultivar than salt sensitive cultivar. Regarding the base of expression in both cultivars, we can infer that from expression matrix of all annotated genes (Table S0 all. gene-expression-annot (for Review)). Moreover, the average FPKM of all genes in each replicate was 33 (Table S0 for review), but the average FPKM of 1518 differentially expressed genes (DEGs) in salt-tolerant and salt-sensitive rice is 48 and 26, respectively (Table S1). From this aspect, the majority of DEGs have higher expression in salt tolerant cultivar than salt sensitive cultivar. Additionally, the main aim of this study is to reveal the differences in response to salt stress between different rice varieties in the same salinity land. Thus, the identified 1518 DEGs including 46 salt-tolerance related genes reported previously can provide references for future salt-tolerance studies.
Discussion
The genes that have been detected in this research should be discussed for their roles, especially the genes that have been characterized by other groups. It can be listed as a table with references.
Answer: The reported genes have been listed in Supplementary Table S2 and cited in the Discussion section.
Minor points
- All the species have to be written with italic letters.
- K+, Na+ should be Na+ and K+.
- Gene names should be written with italic letters, while protein names can be in non-italic letters.
- Figure legends:
4.1 Figure 1 add the scale for red and blue colors in the figure.
4.2 Figure 2, subfigure label, A, B and C should be added to the figure. Full words for abbreviation, such as BP, CC, MF, should be added to the legend.
4.3 Table 3, please provide the meaning of CAS, log2FC and VIP in the table legend.
4.4 The resolution of Figure 4 should be increased.
Answer: Thanks for your suggestion on optimizing the Figures and their legend. We have revised these in the main text and Figures. Thanks again for your suggestions on improving the performance of this manuscript.
Please refer to the attachment for the revised manuscript.

Reviewer 2 Report
See attachment

Overall, English is understandable. However, the sections of Results and Materials and Methods need to be significantly improved. Especially in the M & M section, passive voice and past tense were not used in many places.
Author Response
Response to Reviewer plants-2566860 Comments
Thank you for your valuable feedback and insightful suggestions on our manuscript. We have carefully revised the manuscript based on your recommendations. Below, we provide a point-by-point explanation addressing each of the comments.
- Throughout the manuscript, the authors did not italicize the names of genes, species, and mutants.
Answer: Thanks for your suggestions on correct expression. Already italic letters
- When describing the rice materials used in this study, the authors sometimes used “cultivar”, sometimes used “variety”, and other times used landrace. Please be consistent.
Answer: The rice material has been unified into variety.
- Overall, English is understandable. However, the language in the Results and Methods sections need to be improved. There are many inconsistent uses of tenses.
Overall, English is understandable. However, the sections of Results and Materials and Methods need to be significantly improved. Especially in the M & M section, passive voice and past tense were not used in many places.
Answer: The tense questions in both the Results and Materials and Methods sections have been uniformly revised to past tense or past perfect tense.
- Figure 2 and 4, it is hard to read the font.
Answer: The resolution of Figure 2-4 has been increased to make the fonts more visible.
- In the section of Plant growth, treatment, and harvested sample, the authors indicated months but no information of the year is given;
Answer: The field experiment was performed in 2022, and we have added it to the Methods section.
- In the transcriptome and metabolome data analyses, the authors only paid attention to upregulated DEGs or DAMs, they should also pay attention to those significantly down-regulated DEGs or DAMs.
Answer: Thanks for the insightful comment. Focusing only on upregulated genes does seem a bit bias, and we have added meaningful downregulated genes to the Results and Discussion sections.
- Check all references and make them consistent. A lot of them don’t have page numbers.
Answer: The reference format has been modified.
Minors:
- Line 79, change “Additional” to “Additionally”;
- Line 109, change “receptor kinase receptor-like-kinase” to “receptor-like kinases”;
- Line 120, delete “both”;
- Line 122, change “saline-tolerant” to “salt-tolerant”;
- Line 127, change “downward” to “downregulated”;
- Line 153, delete “the examine”;
- Line 154-155, delete “of TLJIAN and HJING under salt stress”;
- Line 169, change “differential” to “differentially”;
- Line 190, delete “er”;
- Line 208, Figure 4 legend, change “differential accumulation” to “differentially accumulated”;
- Line 217, change “biological” to “biotic”;
- Line 239, change “transport” to “transporter”;
- 248, change “interesting” to “interest”;
- Line 296, change “was” to “were”;
- Line 308, change “green mouth” to “greenhouse”?
- Line 371, change “transcriptomic and metabolomic” to “transcriptomics and metabolomics”;
- Line 385, change “of salt tolerance” to “in salt-tolerant”;
- Line 397, delete “varities”.
Answer: Thanks for your suggestions on correct expression. Vocabulary expressions had been revised in accordance with the reviewers' comments

Reviewer 3 Report
Dear Editors,
I have read 'plants-2566860: Integrated analysis of transcriptome and metabolome reveals molecular mechanisms of rice with different salinity tolerances' carefully. The writing of the paper is standard, and the experiment is reasonable, which meets the publication requirements of Plants. In sum, my evaluation of the article is positive, although I believe some minor points should be addressed before publication.
1. Regarding the last paragraph in the Introduction section, the summary of the results on differential metabolites from lines 91 to 93 can be revised to indicate which specific metabolites are important in the process of salt stress tolerance, therefore strengthening the highlights of this manuscript.
2. Regarding the Results section, Tables such as Table 1 and Table 2 need adding p-value, hence, please review the full text and mark it in the table.
3. Regarding the Figures, the name of them and the cited name in the main text should be consistent. Moreover, the legend of the Figures can be added more detailed information. In detail, Figure.1 lacks a legend, e.g., what does red or blue mean respectively and Figure.2 needs to be labeled (a), (b), (c).
4. Figure4 can be divided in to two figures to display the Correlation between differentially expressed transporter genes, transcription factors and DAMs, respectively. Additionally, the legend for Figure.4 needs to clearly describe what the different colored lines and fonts, and different shaped boxes represent.
5. Regarding the Discussion section, it would be more informative if this paper could include a discussion of the broader applications of the study's findings for crop improvement and identify potential areas for further investigation. Additionally, this section can be contracted to concentrate the main theme of this manuscript.
6. Regarding the Methods, the approach of using MapMan should be described.
Author Response
Response to Reviewer plants-2566860 Comments
Dear reviewer:
Thank you for your valuable feedback and insightful suggestions on our manuscript. We have carefully revised the manuscript based on your recommendations. Below, we provide a point-by-point explanation addressing each of the comments.
- Regarding the last paragraph in the Introduction section, the summary of the results on differential metabolites from lines 91 to 93 can be revised to indicate which specific metabolites are important in the process of salt stress tolerance, therefore strengthening the highlights of this manuscript.
Answer: Thanks for your suggestions on highlighting the key results. Specific salt-tolerant metabolites have been added.
- Regarding the Results section, Tables such as Table 1 and Table 2 need adding p-value, hence, please review the full text and mark it in the table.
Answer: Thanks for your suggestions on improving Tables performance. Table1 and Table 2 p-value corresponding to each gene had been added.
- Regarding the Figures, the name of them and the cited name in the main text should be consistent. Moreover, the legend of the Figures can be added more detailed information. In detail, Figure.1 lacks a legend, e.g., what does red or blue mean respectively and Figure.2 needs to be labeled (a), (b), (c).
Answer: Thanks for your suggestions on enhancing Figures. The name of the charts and graphs have been checked to ensure the consistence with that cited in the main text. For Figure1, legends and icons have been added to it.
- Figure4 can be divided in to two figures to display the Correlation between differentially expressed transporter genes, transcription factors and DAMs, respectively. Additionally, the legend for Figure.4 needs to clearly describe what the different colored lines and fonts, and different shaped boxes represent.
Answer: Thanks for your concerns. Figure 4 has been split into two panels showing the correlations between differentially expressed transporter genes, transcription factors, and DAMs, respectively. Moreover, the legend has been improved.
- Regarding the Discussion section, it would be more informative if this paper could include a discussion of the broader applications of the study's findings for crop improvement and identify potential areas for further investigation. Additionally, this section can be contracted to concentrate the main theme of this manuscript.
Answer: Thanks for your suggestion on optimizing the Discussion section. For example, the literatures of downregulated DEGs and DAMs were added to the discussion section.
- Regarding the Methods, the approach of using MapMan should be described.
Answer: Thanks for your concern. The use of Mapman has been described in the materials and methods section.
Please see the attachment for the revised manuscript.
